# Efficient Bi-Directional Verification of ReLU Networks via Quadratic Programming

## Abstract

Neural networks are known to be sensitive to adversarial perturbations. To investigate this undesired behavior we consider the problem of computing the distance to the decision boundary (DtDB) from a given sample for a deep neural net classifier. In this work we present an iterative procedure where in each step we solve a convex quadratic programming (QP) task. Solving the single initial QP already results in a lower bound on the DtDB and can be used as a robustness certificate of the classifier around a given sample. In contrast to currently known approaches our method also provides upper bounds used as a measure of quality for the certificate. We show that our approach provides better or competitive results in comparison with a wide range of existing techniques.

## 1 Introdution

The high predictive power of neural network classifiers makes them the method of choice to tackle challenging classification problems in many areas. However, questions regarding the robustness of their performance under slight input perturbations still remain open, severely limiting the applicability of deep neural net classifiers to sensitive tasks that require certification of the obtained results.

In recent years this issue gained a lot of attention, resulting in a large variety of methods tackling tasks ranging from adversarial attacks and defenses against these to robustness verification and robust training. In this work we focus on robustness verification of deep classifiers. That is, computing the distance from a given anchor point $x^0$ in the input space to its closest adversarial, i.e. a point that is assigned a different class label by the network. This problem plays a fundamental role in understanding the behavior of deep classifiers and essentially provides the only reliable way to assess classifier robustness. Unfortunately, its complexity class does not allow a polynomial time algorithm. For deep classifiers with ReLU activation the verification problem can equivalently be reformulated as a mixed integer programming (MIP) task and was shown to be NP-complete by Katz et al. (2017). Even worse, Weng et al. (2018) showed that an approximation of the minimum adversarial perturbation of a certain (high) quality cannot be found within polynomial time.

**Related work**   There exist three streams of related work on robustness verification of deep ReLU classifiers. This categorization is based on whether they are solving the verification problem exactly or verifying a bound on the distance to the decision boundary (DtDB).

The first group of methods are **exact verification** approaches. As mentioned above, the verification task can be modeled using MIP techniques. Katz et al. (2017) present a modification of the simplex algorithm that can be used to solve the verification task exactly for smaller ReLU networks based on satisfiable modulo theory (SMT). Other approaches (see Ehlers (2017)) rely on SMT solvers when solving the described task. Bunel et al. (2018) provide an overview and comparison of those. Other exact methods (Dutta et al. (2018); Lomuscio and Maganti (2017); Tjeng et al. (2017)) deploy MIP solvers together with presolving to find a tight formulation of the MIP problem or (Jordan et al. (2018)) use an algorithm to find the largest ball around the anchor point that touches the decision boundary.

The second popular class of methods for verifying classifier robustness deals with **verification of an $\epsilon$-neighborhood**: given an anchor point $x^0$ and an $\epsilon > 0$, the task is to verify whether an adversarial point exists within the $\epsilon$ neighborhood of $x^0$ which is defined with respect to a certain norm in the

input space. All existing methods relax the initial problem and require bounds on activation inputs in each layer. These bounds should be as tight as possible to ensure good final results. Raghunathan et al. (2018a;b); Dvijotham et al. (2018; 2019) consider semidefinite (SDP) and linear (LP) problems as relaxations of the $\epsilon$-verification problem. Wong and Kolter (2018) replace ReLU constraints by linear constraints and consider the dual formulation of the obtained LP relaxation. Weng et al. (2018) present an approach that also uses linear functions (later extended to quadratic functions by Zhang et al. (2018)) to deal with nonlinear activation functions and propagate the layer-wise output bounds until the final layer. Finally Hein and Andriushchenko (2017); Tsuzuku et al. (2018) use the Lipschitz constant of the transformations within classifier's architecture.

Our approach belongs to the third group of approaches dealing with **constructing lower bounds** on DtDB without restricting admissible adversarial points to a given neighborhood. The $\epsilon$-verification task is closely related to the problem of constructing lower bounds on DtDB: each $\epsilon$-neighborhood that can be certified as adversarial-free immediately provides a lower bound on the minimal adversarial perturbation magnitude. It is also a common strategy for the methods from the previous group to use a binary search or a Newton method on top of their algorithm to find the largest $\epsilon$ such that the $\epsilon$-neighborhood around $x^0$ can still be verified as robust. Croce et al. (2019) leverage the piecewise affine nature of the outputs of a ReLU classifier and compute lower bounds on DtDB by assuming that the classifier behaves globally the same way it does in the linear region around the given anchor point.

**Adversarial attacks** Constructing misclassified examples that are close to the anchor point can be considered as a complementary research direction to robustness verification since each adversarial example by definition provides an upper bound on the DtDB. Many methods were proposed to construct such points (Szegedy et al. (2014); Goodfellow et al. (2015); Kurakin et al. (2016); Papernot et al. (2016); Madry et al. (2017); Carlini and Wagner (2017).

**Robust training** The question of how to actually train a robust classifier is closely related to robustness verification since the latter might allow us to construct some type of robust loss based on the insights from the verification procedure (see Hein and Andriushchenko (2017); Madry et al. (2017); Wong and Kolter (2018); Raghunathan et al. (2018a); Tsuzuku et al. (2018); Wang et al. (2018); Croce et al. (2019)). We leave this direction for future work.

**Contributions**

1. We propose a novel **relaxation of the DtDB problem in form of a QP task** allowing efficient computation of high quality lower bounds on the DtDB in $l_2$-norm with an extension to $l_\infty$-norm. For networks with up to two hidden layers we reach state-of-the-art performance with an improvement of over 50% when compared to the bounds obtained from methods based on LP relaxations (CROWN by Zhang et al. (2018) and ConvAdv by Wong and Kolter (2018)). Furthermore, our method performs much faster than methods based on SDP relaxations (Raghunathan et al., 2018b), while providing smaller lower bounds. This is a fundamental property due to the difference in computational complexity between SDP and QP tasks.

2. Unlike $\epsilon$-verification techniques, we provide a lower bound on DtDB **without an initial guess and without computing bounds for the neuron activation values** in each layer. Such bounds have to be tight enough to verify non-trivial neighborhoods and play an important role in other relaxation techniques such as the SDP based approaches by Raghunathan et al. (2018b); Dvijotham et al. (2019).

3. In addition to computing the lower bounds after solving a single QP, **our approach allows to construct adversarial examples by repeatedly choosing a new anchor on the boundary of the verified region** and solving its associated QP. One of the main limitations of competing inexact approaches is the fact that the situation around the given anchor point outside of the certified radius remains unclear. Since these approaches rely on relaxations of the initial problem, failed certification does *not* indicate the presence of an adversarial example. Instead, when using these methods, one must rely on externally constructed adversarial examples to obtain an upper bound. We present **a bi-directional robustness verification technique** that uses the verification mechanism itself to construct upper bounds and assess the quality of the lower bounds.

## 2 NOTATION AND IDEA

We consider a neural network consisting of $L$ linear transformations representing dense, convolutional, skip or average pooling layers and $L - 1$ ReLU activations (no activation after the last hidden layer). The number of neurons in layer $l$ is denoted as $n_l$ for $l = 0, \ldots, L$, meaning that the data has $n_0$ features and $n_L$ classes. Furthermore, we present our analysis for the $l_2$-norm as perturbation magnitude measure since only few available methods are applicable to this setting. To make our method comparable with Raghunathan et al. (2018b) a generalization to $l_\infty$-perturbations is addressed in Appendix A.

Given sample $x^0 \in \mathbb{R}^{n_0}$, weight matrices $W^l \in \mathbb{R}^{n_l \times n_{l-1}}$, and bias vectors $b^l \in \mathbb{R}^{n_l}$, we define the output of the $i$-th neuron in the $l$-th layer after the ReLU activation as

$$x_i^l = \left[ W_i^l x^{l-1} + b_i^l \right]_+ \text{ and} \tag{1}$$
$$f_i(x^0) = x_i^L = W_i^L x^{L-1} + b_i^L$$

where $[x]_+$ is the positive part of $x$ and $f(x^0) = x^L$ denotes the output of the complete forward pass through the network. We start with the observation that the ReLU activation function acts as one of two linear functions depending on its input's sign. That allows us to reformulate it (see Section 3.1) and obtain an optimization problem with so called linear complementarity constraints (also used by Raghunathan et al. (2018b); Dvijotham et al. (2019) for $\epsilon$-verification). Note that for each pair of scalars $a$ and $b$ the following holds.

$$b = [a]_+ \iff b \geq 0, \ b - a \geq 0, \ b(b - a) = 0 \tag{2}$$

## 3 OPTIMIZATION PROBLEM

### 3.1 FORMULATION OF DtDB

For a given sample $\tilde{x}^0$, pre-trained neural net $f$, predicted label $\tilde{y}$ and target label $y$ we aim to find the closest point to $\tilde{x}^0$ in $\mathbb{R}^{n_0}$ that has a larger or equal probability of being classified as $y$ compared to the initial label. This can be done by solving the following optimization problem.

$$\min_{x^0 \in \mathbb{R}^{n_0}} \|x^0 - \tilde{x}^0\|^2, \text{ s.t. } (e_{\tilde{y}} - e_y)^T f(x^0) \leq 0, \tag{DtDB}$$

where $e_i$ is the $i$-th unit vector in $\mathbb{R}^{n_L}$ and $\|x\|$ denotes the Euclidean norm of $x$. Note that we explore the boundary between classes $y$ and $\tilde{y}$ around $\tilde{x}^0$. To compute the distance from $\tilde{x}^0$ to the (full) decision boundary, one needs t compute the solution for all target labels $y = 1, \ldots, n_L$ except $\tilde{y}$.

We can "unfold" the above optimization problem using (1), where $x$ denotes a container with all variables $x^0, \ldots, x^L$.

$$\min_{x \in \mathbb{R}^n} \|x^0 - \tilde{x}^0\|^2, \text{ s.t. } (e_{\tilde{y}} - e_y)^T x^L \leq 0, \quad x^L = W^L x^{L-1} + b^L$$
$$x^l = \text{ReLU}(W^l x^{l-1} + b^l) \text{ for } l = 1, \ldots, L - 1$$

We apply (2) to reformulate the problem and eliminate $x^L$, such that from now on $x$ contains only the remaining variables $x^0, \ldots, x^{L-1}$ and $n = n_0 + \ldots + n_{L-1}$.

$$\min_{x \in \mathbb{R}^n} \|x^0 - \tilde{x}^0\|^2, \text{ s.t. } (e_{\tilde{y}} - e_y)^T \left( W^L x^{L-1} + b^L \right) \leq 0 \tag{DtDB}$$

$$\left( x^l \right)^T \left( x^l - \left( W^l x^{l-1} + b^l \right) \right) = 0 \text{ for } l = 1, \ldots, L - 1 \tag{3}$$
$$x^l - \left( W^l x^{l-1} + b^l \right) \geq 0, \ x^l \geq 0 \text{ for } l = 1, \ldots, L - 1 \tag{4}$$

### 3.2 QP RELAXATION

Next we consider a Lagrangian relaxation of DtDB without constraints (3). We will refer to the resulting problem as QPRel($\lambda$), when the explicit dependency on the multipliers is required.

$$\min_{x \in \mathbb{R}^n} \|x^0 - \tilde{x}^0\|^2 + \sum_{l=1}^{L-1} \lambda_l \left(x^l\right)^T \left(x^l - \left(W^l x^{l-1} + b^l\right)\right), \text{ s.t.} \qquad \text{(QPRel)}$$

$$(e_{\tilde{y}} - e_y)^T \left(W^L x^{L-1} + b^L\right) \leq 0 \qquad (5)$$

$$x^l - \left(W^l x^{l-1} + b^l\right) \geq 0, \ x^l \geq 0 \text{ for } l = 1, \dots, L-1 \qquad (6)$$

The obtained problem is indeed a QP with linear constraints. We need to clarify two questions. How does the problem QPRel help us with solving DtDB and how can this problem itself be solved efficiently?

**QPRel vs. DtDB**  First, we describe properties of the solution of QPRel providing information about DtDB. To do so, we define for arbitrary vectors $x^0 \in \mathbb{R}^{n_0}, \dots, x^{L-1} \in \mathbb{R}^{n_{L-1}}$ and $\lambda \in \mathbb{R}_+^{L-1}$

$$c(x, \lambda) := \sum_{l=1}^{L-1} \lambda_l \left(x^l\right)^T \left(x^l - \left(W^l x^{l-1} + b^l\right)\right) \qquad (7)$$

as the corresponding propagation gap. It follows directly from the definition that for arbitrary non-negative $\lambda$ it holds that:

- if $x$ is feasible for DtDB we have $c(x, \lambda) = 0$, meaning that $x$ equals the vector obtained by propagating $x^0$ through the neural net as defined in (1),
- if $x$ is feasible for QPRel we have $c(x, \lambda) \geq 0$, meaning that there might be a slack between the true output of layer $l$ when getting $x^0$ as an input and the value of $x^l$ obtained by QPRel.

In general the following holds for the relation between the solution of QPRel and DtDB (see also Figure 1, every point in the hatched area is certified to be classified as belonging to the class $\tilde{y}$).

**Lemma 1.** *Denote the solution of QPRel by $x_{qp}$ and the square root of its optimal objective value by $d_{qp}$, let $d$ be the square root of the optimal objective value of DtDB. The following holds:*

1. *$d_{qp} \leq d$ and when $c(x_{qp}, \lambda) = 0$ we have $d_{qp} = d$ and $x_{qp}$ is optimal for DtDB.*

2. *For two non-negative $\lambda^1, \lambda^2$ with $\lambda^1 \leq \lambda^2$ elementwise it holds that $d_{qp}(\lambda^1) \leq d_{qp}(\lambda^2)$.*

The first result from Lemma 1 ensures that $d_{\text{qp}}$ provides a radius of a certified neighborhood around the anchor point. Whereas the second part indicates that we should choose $\lambda$ as large as possible to get our lower bound closer to DtDB. Unfortunately, as we show below, the problem QPRel becomes non-convex for large values of $\lambda$. While one could try to tackle a non-convex QP with proper optimization methods, we will address necessary conditions such that QPRel is guaranteed to be convex and can be solved efficiently next.

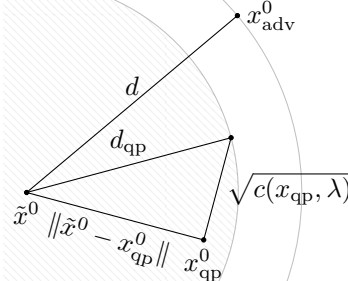

Figure 1: Setting of the optimal solutions for DtDB $x_{\text{adv}}$ and QPRel $x_{\text{qp}}$.

**Convexity of QPRel**  To look into the problem QPRel in more detail we introduce the Hessian $M^\lambda$ (which is a constant matrix) of its objective function.

**Definition 1.** Let $E_l \in \mathbb{R}^{n_l \times n_l}$ be the identity matrix of the corresponding dimension. We define $M^\lambda \in \mathbb{R}^{n \times n}$ as the following symmetric block tridiagonal matrix

$$M^\lambda := \begin{pmatrix} D_0 & M_1^T & & \\ M_1 & D_1 & M_2^T & \\ & M_2 & \ddots & \ddots \\ & & \ddots & D_{L-1} \end{pmatrix} \text{ with } M_l := -\frac{1}{2}\lambda_l W^l, D_l := \lambda_l E_l \text{ and } \lambda_0 := 1.$$

Using this matrix we can rewrite the objective function from QPRel as (see Appendix C, Lemma 2):

$$\min_{x \in \mathbb{R}^n} x^T M^\lambda x + x^T \bar{b}(\lambda, \tilde{x}^0) + \bar{c}(\tilde{x}^0), \text{ s.t. (5), (6)}$$

where $\bar{b}$ and $\bar{c}$ influence only the linear and constant terms and are therefore not relevant in this section. From this reformulation we can clearly see that the matrix $M^\lambda$ determines the (non-)convexity of the objective function. The following theorem provides sufficient and necessary conditions on $\lambda$ depending on the weights $W_l$ assuring that $M^\lambda$ is positive semi-definite. This allows us to use off-the-shelf QP-solvers with excellent convergence properties.

**Theorem 1.** *Let $W^1, \dots, W^{L-1}$ be the weights of an arbitrary pre-trained neural net and $\|W\|$ the spectral norm of an arbitrary matrix. Then the following two conditions for $\lambda$ provide correspondingly a sufficient and a necessary criterion for the matrix $M^\lambda$ to be positive semi-definite.*

$$\text{(suf. condition)} \qquad \lambda_1 \le \frac{2\lambda_0}{\|W^1\|^2} \text{ and } \lambda_l \le \frac{\lambda_{l-1}}{\|W^l\|^2} \qquad \text{for } l = 2, \dots, L-1 \qquad (8)$$

$$\text{(nec. condition)} \qquad \lambda_l \le \frac{4\lambda_{l-1}}{\|W^l\|^2} \qquad \text{for } l = 1, \dots, L-1 \qquad (9)$$

*Furthermore, we define $\underline{\lambda}$ and $\bar{\lambda}$ that correspondingly satisfy conditions (8) and (9) with equality:*

$$\underline{\lambda}_l = 2 \prod_{k=1}^{l} \frac{1}{\|W^k\|^2}, \quad \bar{\lambda}_l = 4^l \prod_{k=1}^{l} \frac{1}{\|W^k\|^2}.$$

*In case with a single hidden layer choosing $\lambda = \bar{\lambda}$ from (9) guarantees $M^\lambda$ to be positive-semi definite.*

We use (8), (9) and our previous results as guidelines for the choice of $\lambda$. Since QPRel($\lambda$) is monotonous in the sense of Lemma 1 we perform a binary search between $\underline{\lambda}$ and $\bar{\lambda}$ from Theorem 1 to find the point closest to $\bar{\lambda}$ such that the QP remains convex. This preprocessing step does not considerably affect the runtime since checking whether a matrix is positive semi-definite is done efficiently by Cholesky decomposition. However, it improves the final bounds by up to a factor two compared to the bounds obtained when using $\lambda = \underline{\lambda}$ without binary search.

Note that this procedure has to be done *once* for a given classifier. The obtained $\lambda$ can subsequently be used to solve QPRel for *all* anchor points and target labels. This is a significant computational advantage compared to SDP based $\epsilon$-verification procedures. For example, the method by Dvijotham et al. (2019) includes the dual multipliers as variables in the SDP problem that has to be solved for each combination of the anchor point, target label and verified epsilon.

## 4 UPPER BOUNDS

Now we describe an iterative procedure generating a sequence of points in $\mathbb{R}^{n_0}$ that converges to a point on the decision boundary. This procedure provides an upper bound on the distance to DtDB and thus allows for bi-directional verification of a classifier.

---
**Algorithm 1:** Algorithm to construct an adversarial example by iteratively solving QPRel.

**Data:** $\tilde{x}^0$, trained neural net and $\lambda$ such that $M^\lambda$ is positive semi-definite, $c_{\text{tol}}$, $\delta$.
**Result:** $d_{\text{ub}}, x_{\text{adv}}^0$

1 **begin**
2     $x^0 \longleftarrow \tilde{x}^0; \ c \longleftarrow +\infty$
3     **while** $c > c_{tol}, pred(\tilde{x}^0) = pred(x^0)$ **do**
4        Solve QPRel($\lambda$) from the anchor point $x^0$:
5        $x_{\text{qp}}, d_{\text{qp}} \longleftarrow$ optimal solution and square root of the objective function value
6        $c \longleftarrow c(x_{\text{qp}}, \lambda)$
7        Choose the next anchor point on the boundary of the current certified region:
8        **if** $c > c_{\text{tol}}$ **then** $x^0 \longleftarrow x^0 + d_{\text{qp}} \frac{x_{\text{qp}}^0 - x^0}{\|x_{\text{qp}}^0 - x^0\|}$ **else** $x^0 \longleftarrow x^0 + d_{\text{qp}}(1+\delta) \frac{x_{\text{qp}}^0 - x^0}{\|x_{\text{qp}}^0 - x^0\|}$
9     **end**
10     $x_{\text{adv}}^0 \longleftarrow x^0; \ d_{\text{ub}} \longleftarrow \|\tilde{x}^0 - x_{\text{adv}}^0\|$
11 **end**

---

**Idea** In each step we verify a certain neighborhood around the current anchor point $x^0$ and then expand the verified region further. For that we choose the next anchor point as follows. **1) Choice of direction:** we go from $x^0$ towards the solution of QPRel since, if the QP relaxation is tight, its solution should be close to an optimal solution of DtDB which is the closest adversarial point to $x^0$. **2) Choice of step size:** we know that there are no adversarial points within the ball of the verified radius $d$ around the anchor, so every step size smaller than $d$ would be unnecessary small. On the other hand, if we proceed with a new anchor point that is strictly farther away than $d$, we might miss an adversarial point lying close to the boundary of the $d$-ball around $x^0$. Therefore, we choose the next anchor point to be on the boundary of the currently verified region, so that every $\epsilon$-ball that we manage to verify around the new point would add to the overall robust set. Algorithm 1 provides a pseudo-code for this procedure.

**Termination** The algorithm terminates as soon as the propagation gap $c(x, \lambda)$ (see (7)) becomes small enough or the anchor point gets misclassified. Note that $c(x, \lambda) = 0$ means that the solution $x$ provides the optimal objective function value of the DtDB problem (see Lemma 1) and, thus, an adversarial example. However, the termination condition $c \leq c_{\text{tol}}$ from Algorithm 1, line 3 cannot ensure that the optimal point $x^0_{\text{qp}}$ from the last iteration belongs to a different class. Therefore, if we stop with $c \leq c_{\text{tol}}$ and the second termination condition is not satisfied, we take an additional step on the boundary of the ball of radius $d_{\text{qp}}(1 + \delta)$, where $d_{\text{qp}}$ is the verified radius and $\delta$ is a tiny offset.

Additionally, we check in each step whether the next anchor point is already misclassified before the condition $c \leq c_{\text{tol}}$ is reached (this can happen in a multi-class setting; and is indeed observed frequently). We empirically verified that all points obtained this way are indeed true adversarials. This means that the sequence of anchor points converges *towards the boundary* and then, if no adversarial point was found yet, makes a step *across the boundary* using a positive, small $\delta$. Note, while the most powerful attacks as proposed by Carlini and Wagner (2017) were empirically shown to succeed for every considered sample, their success in not guaranteed and might require tuning of hyper-parameters. In contrast, we provide a formal proof of our method's convergence below.

Figure 2 illustrates how the constructed sequence of anchor points and solutions of QPRel might look like for the setting of Figure 1. Here only the first three steps are shown assuming that we would reach either a small value for the propagation gap or an adversarial point in the last step. Blue arrows show how we proceed from the solution of QPRel to the next anchor point in each iteration (compare to the update formula from Algorithm 1, line 8).

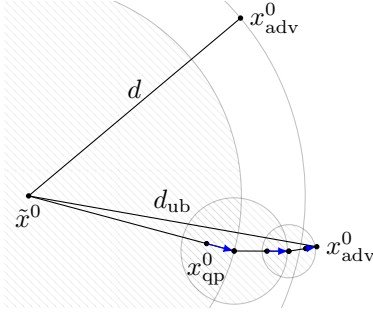

**Convergence** The following result holds for the convergence rate of Algorithm 1 under the assumption that the propagation gap $c(x, \lambda)$ is not too large with respect to the optimal objective function value for the QPRel problem solved in each step.

Figure 2: Three steps of Algorithm 1.

**Theorem 2.** *For $k = 0, 1, \ldots$ let $c_k$ be the propagation gap as defined in (7) and $d_k^2$ the optimal objective function value we obtain after solving QPRel in the k'th iteration of Algorithm 1. Assume that there exists $\alpha \in [0, 1]$ such that for all $k$ the following holds.*

$$\frac{c_k}{d_k^2} \leq 1 - \alpha^2 \text{ or equivalently } \frac{\|x^0 - x^0_{qp}\|}{d_k} \geq \alpha \tag{10}$$

*where $x^0$ is the anchor point and $x_{qp}$ is the optimal solution of QPRel in iteration $k$ (we omit the iteration index $k$ on $x$'s). Then for all $k$*

$$d_{k+1}^2 \leq 2(1 - \alpha)d_k^2, \quad c_k \leq (2(1 - \alpha))^{k+1}d_0^2. \tag{11}$$

*Furthermore, if $\alpha > \frac{1}{2}$ then the number of iterations $\bar{k}$ such that Algorithm 1 terminates with $c_{\bar{k}} \leq c_{tol}$ is bounded by*

$$\bar{k} \leq \frac{\log c_{tol} - 2 \log d_0}{\log 2(1 - \alpha)} - 1. \tag{12}$$

First of all, this result ensures that Algorithm 1 terminates after a final number of steps if (10) holds for some $\alpha \in (\frac{1}{2}, 1]$. Moreover, it provides a hint about how many iterations we would have to do if we decreased the threshold $c_{\text{tol}}$ by a factor of 10. In this case the upper bound (12) on the number of iterations would increase by $\Delta k_\alpha = -(\log 2(1-\alpha))^{-1}$ with $\Delta k_\alpha \to \infty$ when $\alpha \to \frac{1}{2}$ and $\Delta k_\alpha \to 0$ when $\alpha \to 1$. Note that $\alpha$ being almost 1 means that $c_k$ is close to 0. In this regime the relaxation is exact, as discussed in Lemma 1, part 1. Therefore, considering $\alpha$ as a measure of tightness of the relaxation QPRel we can conclude that, when the relaxation is tight (i.e. $\alpha \approx 1$), decreasing $c_{\text{tol}}$ by an order of magnitude will not be as costly as when $\alpha$ is smaller.

**Related work**    In relation to the methodology proposed in this section we consider the work on exact DtDB computation by Jordan et al. (2018). The proposed algorithm GeoCert iteratively computes the radius of the largest $l_p$-ball that stays within a certain polyhedral complex. The latter is incrementally updated in each step until the decision boundary is reached. This way one get a sequence of lower bounds on DtDB until the true distance is found. GeoCert is significantly different to our method: 1.) It does not employ a relaxation of (DtDB). 2.) It does not change the anchor point, but rather expands the considered polyhedral complices by analyzing the linear regions of a ReLU classifier. 3.) While it can use an upper bound to speed up the algorithm, the complexity of computing its own exact distance is comparable to MIP-based approaches as shown in Jordan et al. (2018).

## 5 EXPERIMENTS

### 5.1 SETUP

The values of the parameters we used as well as the runtime of the compared algorithms is shown in Appendix B.1 and B.2. To solve the QP tasks we use Gurobi Optimization (2018).

**Datasets and classifiers**    The experiments are performed using the MNIST and Fashion-MNIST datasets scaled such that the feature values lie in $[0, 1]$ interval. For each of the datasets we use the correctly classified samples from 1200 train points (label distribution preserved) to evaluate the verification approaches. The exact number of used samples is reported for each experiment in Appendix D.

For classification we take ReLU networks consisting of dense linear layers. The architectures we used are named by the number of hidden layers and the number of neurons per hidden layer: **L1N50**, **L2N50**, **L3N50**, **L4N50**, **L5N50** and **L1N300** with the same number of neurons in each hidden layer. For each architecture we use normally trained classifiers as well as robustly trained ones (indicated by suffix **R**, e.g. **L1N50R**) using the method by Wong and Kolter (2018) with $\epsilon = 1.58$ in $l_2$-setting. The weights will be available in the project repository.

**Competitors**    We compare our approach QPRel with the following $\epsilon$-verification methods: ConvAdv by Wong and Kolter (2018) based on the LP relaxation of ReLU constraints (we use its implementation supporting the $l_2$-norm by Croce et al. (2019)), CROWN by Zhang et al. (2018) which is a layerwise bound propagation technique including performance boosting quadratic approximations and warm start (for each setting we report only the best result from these two competitors), and SDPRel by Raghunathan et al. (2018b) based on a SDP relaxation solved by MOSEK. Finally, we take the iterative PGD attack with 200 steps and a step size of $2.5\epsilon/200$ (in accordance with Madry et al. (2017)) as a competitor for QPRel-UB. We use the notation QPRel-LB or QPRel-UB, when we want to emphasize what bounds are meant.

### 5.2 RESULTS

Results on MNIST in $l_2$-setting are shown in Table 1. We include the complete results in Appendix D. We run all methods for each of the considered samples and report the following metrics.

*AvgBound*: The average value of the bounds obtained from QPRel and the corresponding competitor. Both lower and upper bounds are reported separately.

***MedRelDiff to QPRel***: The median of the *relative* difference between the bounds (e.g. QPRel-LB minus CROWN and then divided by CROWN). *Positive* values for the *lower* bounds mean our bounds are better in average over the samples; *negative* values are better for the *upper* bounds.

$\epsilon$ **to hit 50% LB-verified**: The number of samples with an adversarial-free radius of $\epsilon$ is monotonically decreasing in $\epsilon$. Therefore, to assess the performance of a verification procedure like QPRel-LB or CROWN we report the smallest $\epsilon$ such that exactly 50% of the samples can be verified. The larger this value, the better.

***VerRatio worst***: While the previous metric only considers the lower bound, we can consider both bounds (LB, UB) jointly by evaluation the *verification ratio*: For a given $\epsilon$, it corresponds to the fraction of samples for which we can either prove that the $\epsilon$-neighborhood is adversarial-free or contains a misclassified point in it. In short: either the lower bound is larger then $\epsilon$ or the upper bound is smaller then $\epsilon$.
Note, that for very small $\epsilon$ values almost all samples will be verified as $\epsilon$-robust by any algorithm providing non-trivial lower bounds. On the other side, when $\epsilon$ is large, attacks will be able to find an adversarial that is even closer to the anchor and verify almost every sample as $\epsilon$-non-robust. Between these two extremes, the verification ratio drops to its minimum for a certain value of $\epsilon_0$. In this setting, the considered algorithm is able to verify the *smallest fraction of samples* – that is, it corresponds to the worst case performance of the bi-directional verification method. We report this smallest fraction in the table, denoted with *VerRatio worst*. The larger, the better.

***MedRelDiff UB-LB***: Lastly, we report the median of the relative difference between the different bounds for each approach (QPRel-UB vs. QPRel-LB or PGD vs. CROWN/ConvAdv). It shows the *tightness* of the certificates when compared to the upper bounds. The smaller, the better.

Table 1: Better bounds and verification ratio for the compact networks ($l_2$-perturbations)

| **MNIST**, $l_2$ | | AvgBound | | MedRelDiff to QPRel (%) | | $\epsilon$ to hit 50% LB-verified | VerRatio worst (%) | MedRelDiff UB-LB (%) |
|---|---|---|---|---|---|---|---|---|
| L/N/R | Method | LB | UB | LB | UB | | | |
| 1/50 | QPRel | **0.42** | **0.49** | – | – | **0.398** | **87.5** | **13.7** |
| 1/50 | CROWN+PGD | 0.25 | 0.72 | +63.0 | -30.8 | 0.239 | 31.4 | 168.7 |
| 1/50/R | QPRel | **1.69** | **1.84** | – | – | **1.698** | **90.3** | **5.3** |
| 1/50/R | CROWN+PGD | 1.52 | 2.68 | +12.8 | -30.8 | 1.567 | 49.8 | 72.0 |
| 2/50 | QPRel | **1.20** | **1.21** | – | – | **1.174** | **99.3** | **<0.01** |
| 2/50 | ConvAdv+PGD | 0.89 | 1.80 | +30.1 | -25.0 | 0.891 | 51.2 | 89.8 |
| 2/50/R | QPRel | 1.39 | **1.81** | – | – | 1.388 | **74.8** | **25.3** |
| 2/50/R | ConvAdv+PGD | **1.47** | 2.62 | -2.9 | -30.8 | **1.527** | 52.4 | 76.3 |
| 3/50 | QPRel | 0.18 | **0.55** | – | – | 0.181 | **26.0** | **200.2** |
| 3/50 | ConvAdv+PGD | **0.19** | 0.71 | -9.3 | -18.2 | **0.199** | 19.9 | 246.4 |
| 3/50/R | QPRel | 1.10 | **1.78** | – | – | 1.091 | **59.5** | **57.7** |
| 3/50/R | ConvAdv+PGD | **1.43** | 2.55 | -21.9 | -30.8 | **1.488** | 51.1 | 74.3 |
| 4/50 | QPRel | 0.09 | **0.58** | – | – | 0.106 | 9.1 | 484.6 |
| 4/50 | ConvAdv+PGD | **0.19** | 0.70 | -50.4 | -18.2 | **0.191** | **19.5** | **255.4** |
| 4/50/R | QPRel | 0.69 | **1.76** | – | – | 0.698 | 35.7 | 142.5 |
| 4/50/R | ConvAdv+PGD | **1.41** | 2.51 | -49.8 | -25.0 | **1.497** | **51.2** | **71.6** |
| 5/50 | QPRel | 0.02 | **0.42** | – | – | 0.037 | 9.5 | 1370.6 |
| 5/50 | ConvAdv+PGD | **0.16** | 0.63 | -84.4 | -35.7 | **0.167** | **19.5** | **274.2** |
| 5/50/R | QPRel | 0.37 | **1.76** | – | – | 0.391 | 20.0 | 357.0 |
| 5/50/R | ConvAdv+PGD | **1.41** | 2.58 | -73.1 | -30.8 | **1.466** | **49.3** | **79.9** |
| 1/300 | QPRel | **0.26** | **0.28** | – | – | **0.243** | **94.0** | **5.5** |
| 1/300 | CROWN+PGD | 0.10 | 0.53 | +149.1 | -43.8 | 0.101 | 17.2 | 385.2 |
| 1/300/R | QPRel | 0.78 | **1.81** | – | – | 0.782 | 42.8 | 120.4 |
| 1/300/R | CROWN+PGD | **1.43** | 2.76 | -44.1 | -30.8 | **1.418** | **46.6** | **83.5** |

**State-of-the-art bounds** For the normally trained networks with a smaller number of hidden layers the lower bounds computed by QPRel are tighter in comparison to the competitors in average and for most individual images (see Table 1, *AvgBound* and *MedRelDiff*). This results in larger values of $\epsilon$ *to hit 50% VerRatio* as well. It seems that the competitors tend to underestimate robustness of these networks. For example, for 50% of the MNIST images classified with **L1N300** we have improved the lower bound from CROWN by over 149%. This discrepancy in the quality of the lower bounds decreases with network's depth and if it was trained using the robust training approach by Wong and Kolter (2018) so that the competitors outperform QPRel's lower bound for robustly trained deep networks.

Noteworthy, the upper bounds derived via QPRel always outperform the one computed by the competing methods – independent of the neural network architecture and the training procedure. That is, the 200-step PGD attack was not able to construct adversarial examples that are closer to the anchor point then those found by QPRel-UB as described in Section 4 in all settings (see Table 1, columns *AvgBound/UB* and *MedRelDiff to QPRel/UB*). Therefore, in the settings where the bounds from QPRel-LB are also tighter as the competitors' we get a much higher *VerRatio worst* value and smaller *MedRelDiff UB-LB* that ensure the good quality of QPRel-LB bounds without computing the DtDB exactly. For example, for **L2N50** the gap between QPRel-LB and QPRel-UB vanishes resulting in *VerRatio worst* of over 99%. That means the computed lower bound is almost the exact DtDB for the majority of the samples. Note that no exact method computing DtDB via MIP techniques is used to achieve that. For larger networks we observe a decay of *VerRatio worst* and looser *MedRelDiff UB-LB*.

**Comparison with SDP-relaxations in $l_\infty$-setting** In order to compare our method with Raghunathan et al. (2018b) we generalize QPRel to the $l_\infty$-setting as described in Appendix A. Note, that the resulting relaxation is looser then the initial QPRel for the $l_2$-setting since we introduce an additional penalty term to make the problem convex. That leads to worse results shown in Tables 9 and 10. To compute the largest $\epsilon$ such that the SDP verification succeeds we perform a binary search between the lower bounds $d_{\mathrm{qp}}$ computed by QPRel-LB and $d_{\max} = 1.0$ which is the maximal perturbation for the $l_\infty$-norm on images. Since this approach takes longer to run we test it only on the **L1N50R** net trained with $\epsilon = 0.1$ (MNIST test data). Further, we speed up this approach by modifying MOSEK parameters (see Table 6 for details) such that the optimization procedure terminates earlier (approximately after a half of the usual number of iterations). We can still rely on the obtained results since we are not interested in the exact value of the SDP objective, but only whether it is positive or negative which was observed to be determined far sooner during the solution process then when the solver would reach a true optimum.

While QPRel-LB in this setting is able to provide tighter bounds then the LP-based approaches for larger computational cost, our bounds are about 1.54 times larger then the ones of SDPRel (see Table 10) – though computed three orders of magnitude faster (see Appendix B.2). This shows that the QP relaxation is less suited then SDPRel for obtaining tight bounds in $l_\infty$-setting as already indicated by the arguments above and in Appendix A but trades this off by much better efficiency.

## 6 CONCLUSION

In this work we present a novel approach to solve the problem of approximating the minimal adversarial perturbations for ReLU networks based in a convex QP relaxation of DtDB. We show that the lower bounds computed with QPRel improve the results of the available LP based methods and allow certification of larger neighborhoods. Since convexity of the underlying QP determines computational efficiency of our approach we derive the necessary and sufficient conditions on the Lagrangian multipliers for it. Additionally, a method that constructs a sequence of points converging to the point on the decision boundary provides us upper bounds on DtDB. Quality of the obtained bounds in the $l_2$-setting is shown to be good enough to verify consistently over 80% of the considered samples for MNIST and FashionMNIST data and simple networks for all values of the perturbation magnitude, while competitors' verification ratio might drop below 20%.

With our contribution we make a step towards robustness verification of deep ReLU-based classifiers. To be able to apply the approach on a wider class of networks it should be generalized to popular architectures beyond ReLU activations and linear layers. Moreover, for deeper networks the proposed

relaxation was shown to provide looser bounds than the other methods indicating an important direction of the future work. Furthermore, there are several points about Theorem 2 that will be addressed in our future work. It is an open question how to verify (10) a priori. Experiments show that for the current choice of $\lambda$ (see the discussion after Theorem 1) this condition does hold for small networks, but not for the larger ones.

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

## A  GENERALIZATION TO $l_\infty$

For comparison with Raghunathan et al. (2018b) we show how our method can be applied to compute bounds on the distance to the closest adversarial measured using the $l_\infty$ norm. A straight forward way would be to modify the objective function accordingly. By introducing a new variable $m$ representing $\|x^0 - \tilde{x}^0\|_\infty^2 = \max_i |x_i^0 - \tilde{x}_i^0|^2$ and $n_0$ new quadratic constraints we get the following versions of QPRel. Note that the quadratic constraints do not harm the complexity since they describe a convex cone and can be handled by the QP-solvers including *gurobi*.

$$\min_{x \in \mathbb{R}^n, \, m \in \mathbb{R}} m + c(x, \lambda), \text{ s.t.}$$
$$(x_i^0 - \tilde{x}_i^0)^2 \le m, \ i = 1, \dots, n_0$$
$$(e_{\tilde{y}} - e_y)^T \left( W^L x^{L-1} + b^L \right) \le 0$$
$$x^l - \left( W^l x^{l-1} + b^l \right) \ge 0, \ x^l \ge 0$$

While this formulation is of a similar structure as the QPRel (quadratic objective and linear plus quadratic constraints), the Hessian of the objective function is not positive semi-definite for any value of $\lambda$. Since $c(x, \lambda)$ is the only source of quadratic terms now (squared distance to the anchor point is now replaced by $m$), the new $M^\lambda$ is of the same form as in Lemma 2, but with $\lambda_0 = 0$. To see that we cannot affect the convexity of the objective function by the parameter $\lambda$ anymore consider vector $x$ with an arbitrary $x^0 \in \mathbb{R}^{n_0}$, $x^1 = \alpha W^1 x^0$ with $0 < \alpha < 1$ and $x^l = 0$ for $l > 1$. Then

$$x^T M^\lambda x = \lambda_1 \left( \|x^1\|^2 - \left( x^1 \right)^T W^1 x^0 \right) = \lambda_1 (\alpha^2 - \alpha) \|W^1 x^0\|^2 < 0$$

meaning that $M^\lambda$ cannot be positive semi-definite.

To overcome this issue we utilize the new quadratic constraints. We return back to a convex QP by considering the following problem with a positive $\mu$.

$$\min_{x \in \mathbb{R}^n, \, m \in \mathbb{R}} m + c(x, \lambda) + \mu \sum_{i=1}^{n_0} \left( (x_i^0 - \tilde{x}_i^0)^2 - m \right), \text{ with}$$
$$(x_i^0 - \tilde{x}_i^0)^2 \le m, \ i = 1, \dots, n_0$$
$$(e_{\tilde{y}} - e_y)^T \left( W^L x^{L-1} + b^L \right) \le 0$$
$$x^l - \left( W^l x^{l-1} + b^l \right) \ge 0, \ x^l \ge 0$$

Clearly, for $0 < \mu \le n_0^{-1}$ the solution of this problem is a finite lower bound on DtDB with the $l_\infty$-norm. On the other side we are back in the setting of Theorem 1 with $\lambda_0 = \mu$ allowing us to use the same framework as before. Results in the $l_\infty$-setting were obtained by solving this problem with $\mu = (2n_0)^{-1}$.

## B  EXPERIMENTS SETUP AND RUNTIME

### B.1  SETUP

$\lambda$ is chosen for each classifier according to Theorem 1 and the discussion afterwards such that an accuracy level of $10^{-4}$ is achieved during the binary search in each $\lambda_l$. For the values of other parameters in Algorithm 1 we choose for all tests $c_{\text{tol}} = 10^{-4}$ and $\delta = 10^{-4}$. Other methods are tested with the default settings as provided in the corresponding repositories.

### B.2  RUNTIME AND NUMBER OF ITERATIONS

Tables 2, 3, 4 and 5 show the average runtime and its standard deviation for networks L1N50R, L1N100, L1N300 and L2N100 as well as the number of iterations Algorithm 1 needs to terminate. During the binary search procedure we apply with SDPRel we always make 10 bisection steps. All tasks necessary for the computation of bounds on DtDB for one sample are run on a single CPU (including the solution of QPs and SDPs with Gurobi Optimization (2018) and MOSEK respectively).

Here, we do not consider ConvAdv as CROWN performs always faster. From the comparison of the time for the computation of our lower bounds (column *Runtime-LB (s)*) and the overall runtime (column *Runtime (s)*) we conclude that obtaining the upper bounds takes $n - 1$ times longer than solving the initial QP where $n$ is approximately the overall number of the iterations in Algorithm 1. While in almost all considered settings $n$ remains smaller than 4 it might become a limiting factor when applying QPRel-UB on larger networks. That means further investigation of the convergence properties of Algorithm 1 is necessary for generalizing the described method. From the comparison of QPRel-LB and CROWN we see the clear advantage of the latter since it doesn't involve any optimization task. However, this advantage comes in cost of the verification properties as discussed above. On the other hand, SDPRel with a binary search provides better bounds, but is about 2000 times slower then QPRel-LB (see the last line in the MNIST section in Table 5).

Table 2: Runtime comparison, train data, $l_2$-setting

| Setting | | | | Runtime-LB (s) | | Runtime-Full (s) | | Nr. iterations(-UB) | |
|---------|-----|------|--------|------|------|------|------|------|------|
| Data | L/N | NrPts | Method | mean | std | mean | std | mean | std |
| MNIST | 1/100 | 1086 | QPRel | 1.467 | 0.142 | 4.200 | 2.091 | 2.845 | 1.416 |
| | 1/100 | 1086 | CROWN | 0.021 | 0.002 | – | – | – | – |
| | 2/100 | 965 | QPRel | 0.280 | 0.017 | 0.472 | 0.140 | 1.679 | 0.484 |
| | 2/100 | 965 | CROWN | 0.034 | 0.013 | – | – | – | – |
| | 1/300 | 1142 | QPRel | 5.421 | 0.309 | 12.631 | 3.414 | 2.317 | 0.629 |
| | 1/300 | 1142 | CROWN | 0.064 | 0.016 | – | – | – | – |
| | 1/50 | 1180 | QPRel | 0.253 | 0.018 | 0.805 | 0.257 | 3.176 | 1.002 |
| | 1/50 | 1180 | CROWN | 0.017 | 0.006 | – | – | – | – |
| F-MNIST | 1/100 | 1083 | QPRel | 1.441 | 0.156 | 4.534 | 4.124 | 3.142 | 2.687 |
| | 1/100 | 1083 | CROWN | 0.025 | 0.009 | – | – | – | – |
| | 2/100 | 924 | QPRel | 0.289 | 0.019 | 0.992 | 0.469 | 3.431 | 1.600 |
| | 2/100 | 924 | CROWN | 0.031 | 0.012 | – | – | – | – |
| | 1/300 | 1094 | QPRel | 5.385 | 0.317 | 18.782 | 12.443 | 3.503 | 2.313 |
| | 1/300 | 1094 | CROWN | 0.063 | 0.018 | – | – | – | – |

Table 3: Runtime comparison, test data, $l_2$-setting

| Setting | | | | Runtime-LB (s) | | Runtime-Full (s) | | Nr. iterations(-UB) | |
|---------|-----|------|--------|------|------|------|------|------|------|
| Data | L/N | NrPts | Method | mean | std | mean | std | mean | std |
| MNIST | 1/100 | 1137 | QPRel | 1.451 | 0.142 | 4.225 | 1.994 | 2.898 | 1.342 |
| | 1/100 | 1137 | CROWN | 0.024 | 0.009 | – | – | – | – |
| | 2/100 | 1028 | QPRel | 0.280 | 0.018 | 0.473 | 0.141 | 1.683 | 0.486 |
| | 2/100 | 1028 | CROWN | 0.033 | 0.012 | – | – | – | – |
| | 1/300 | 1166 | QPRel | 5.353 | 0.328 | 12.954 | 3.386 | 2.398 | 0.619 |
| | 1/300 | 1166 | CROWN | 0.059 | 0.017 | – | – | – | – |
| | 1/50 | 1212 | QPRel | 0.257 | 0.018 | 0.817 | 0.256 | 3.183 | 0.993 |
| | 1/50 | 1212 | CROWN | 0.016 | 0.006 | – | – | – | – |
| F-MNIST | 1/100 | 1061 | QPRel | 1.456 | 0.148 | 4.635 | 4.186 | 3.161 | 2.676 |
| | 1/100 | 1061 | CROWN | 0.024 | 0.008 | – | – | – | – |
| | 2/100 | 949 | QPRel | 0.276 | 0.022 | 0.985 | 0.455 | 3.557 | 1.613 |
| | 2/100 | 949 | CROWN | 0.031 | 0.012 | – | – | – | – |
| | 1/300 | 1069 | QPRel | 5.406 | 0.301 | 19.585 | 12.080 | 3.638 | 2.251 |
| | 1/300 | 1069 | CROWN | 0.058 | 0.016 | – | – | – | – |

Table 4: Runtime comparison, train data, $l_\infty$-setting

| Setting | | | | Runtime-LB (s) | | Runtime-Full (s) | | Nr. iterations(-UB) | |
| --- | --- | --- | --- | --- | --- | --- | --- | --- | --- |
| Data | L/N | NrPts | Method | mean | std | mean | std | mean | std |
| MNIST | 1/100 | 1086 | QPRel | 6.202 | 1.283 | 19.104 | 63.168 | 2.752 | 8.231 |
| | 1/100 | 1086 | CROWN | 0.019 | 0.005 | – | – | – | – |
| | 2/100 | 965 | QPRel | 1.392 | 0.143 | 4.110 | 1.738 | 2.988 | 1.293 |
| | 2/100 | 965 | CROWN | 0.029 | 0.011 | – | – | – | – |
| | 1/300 | 1142 | QPRel | 10.470 | 0.854 | 14.567 | 7.457 | 1.400 | 0.732 |
| | 1/300 | 1142 | CROWN | 0.054 | 0.017 | – | – | – | – |
| | 1/50 | 1180 | QPRel | 2.634 | 0.916 | 16.813 | 19.141 | 5.818 | 3.660 |
| | 1/50 | 1180 | CROWN | 0.016 | 0.006 | – | – | – | – |
| F-MNIST | 1/100 | 1083 | QPRel | 5.750 | 1.314 | 18.447 | 66.807 | 2.651 | 7.627 |
| | 1/100 | 1083 | CROWN | 0.018 | 0.004 | – | – | – | – |
| | 2/100 | 924 | QPRel | 1.500 | 0.179 | 3.496 | 2.978 | 2.348 | 1.954 |
| | 2/100 | 924 | CROWN | 0.026 | 0.010 | – | – | – | – |
| | 1/300 | 1094 | QPRel | 10.636 | 0.882 | 11.541 | 4.031 | 1.085 | 0.379 |
| | 1/300 | 1094 | CROWN | 0.053 | 0.017 | – | – | – | – |

Table 5: Runtime comparison, test data, $l_\infty$-setting

| Setting | | | | Runtime-LB (s) | | Runtime-Full (s) | | Nr. iterations(-UB) | |
| --- | --- | --- | --- | --- | --- | --- | --- | --- | --- |
| Data | L/N | NrPts | Method | mean | std | mean | std | mean | std |
| MNIST | 1/100 | 1137 | QPRel | 6.208 | 1.250 | 22.898 | 75.986 | 3.145 | 8.924 |
| | 1/100 | 1137 | CROWN | 0.020 | 0.007 | – | – | – | – |
| | 2/100 | 1028 | QPRel | 1.321 | 0.145 | 3.912 | 1.619 | 3.002 | 1.266 |
| | 2/100 | 1028 | CROWN | 0.029 | 0.011 | – | – | – | – |
| | 1/300 | 1166 | QPRel | 10.512 | 0.844 | 15.852 | 8.343 | 1.515 | 0.808 |
| | 1/300 | 1166 | CROWN | 0.048 | 0.014 | – | – | – | – |
| | 1/50 | 1054 | QPRel | 2.683 | 1.041 | 17.439 | 16.539 | 6.102 | 3.307 |
| | 1/50 | 1054 | CROWN | 0.016 | 0.006 | – | – | – | – |
| | 1/50 | 1054 | SDPRel | 4338.017 | 949.572 | – | – | 10 | 0.000 |
| F-MNIST | 1/100 | 1061 | QPRel | 5.805 | 1.193 | 21.649 | 73.836 | 3.107 | 8.900 |
| | 1/100 | 1061 | CROWN | 0.020 | 0.007 | – | – | – | – |
| | 2/100 | 949 | QPRel | 1.454 | 0.164 | 3.407 | 2.689 | 2.352 | 1.827 |
| | 2/100 | 949 | CROWN | 0.025 | 0.009 | – | – | – | – |
| | 1/300 | 1069 | QPRel | 10.686 | 0.927 | 11.829 | 5.051 | 1.107 | 0.466 |
| | 1/300 | 1069 | CROWN | 0.048 | 0.014 | – | – | – | – |

## C    PROOFS

**Lemma 1.** *Denote the solution of QPRel by $x_{qp}$ and the square root of its optimal objective value by $d_{qp}$, let $d$ be the square root of the optimal objective value of DtDB. The following holds:*

1. *$d_{qp} \leq d$ and when $c(x, \lambda) = 0$ we have $d_{qp} = d$ and $x$ is optimal for DtDB.*

2. *For two non-negative $\lambda^1, \lambda^2$ with $\lambda^1 \leq \lambda^2$ elementwise it holds that $d_{qp}(\lambda^1) \leq d_{qp}(\lambda^2)$.*

*Proof.* Assume $x_{\text{adv}}$ is the optimal solution of DtDB. Then it is an admissible point of QPRel as well and $c(x_{\text{adv}}, \lambda) = 0$ since $x_{\text{adv}}^l = W^l x_{\text{adv}}^{l-1}$ for $l = 1, \ldots, L-1$. Since $x_{\text{qp}}$ is optimal for QPRel and $x_{\text{adv}}$ is just its admissible point we get that

$$d^2 = \|x_{\text{adv}}^0 - \tilde{x}^0\|^2 = \|x_{\text{adv}}^0 - \tilde{x}^0\|^2 + c(x_{\text{adv}}, \lambda) \geq \|x_{\text{qp}}^0 - \tilde{x}^0\|^2 + c(x_{\text{qp}}, \lambda) = d_{\text{qp}}^2$$

proving the first claim. The second one follows from the fact that $c(x, \lambda)$ for a given $x$ is a linear function of $\lambda$:

$$c(x, \lambda) = \lambda^T \begin{pmatrix} c(x, e_1) \\ \vdots \\ c(x, e_{L-1}) \end{pmatrix}$$

where each $c(x, e_l) = \left(x^l\right)^T \left(x^l - \left(W^l x^{l-1} + b^l\right)\right)$ is non-negative for admissible $x$ because of the non-negativity constraints (4). Therefore the claim follows immediately from the assumption that $\lambda_l^1 \leq \lambda_l^2$ for all $l$

$$c(x, \lambda^1) = \sum_{l=1}^{L-1} \lambda_l^1 c(x, e_l) \leq \sum_{l=1}^{L-1} \lambda_l^2 c(x, e_l) = c(x, \lambda^2).$$

$\square$

**Lemma 2.** *Objective function of QPRel can be reformulated as*

$$d_{qp}(\lambda, x) = x^T M^\lambda x + x^T \bar{b}(\lambda, \tilde{x}^0) + \bar{c}(\tilde{x}^0)$$

*where terms $\bar{b}$ and $\bar{c}$ don't depend on $x$.*

*Proof.* The proof is done by sorting the quadratic, linear and constant terms in the objective function of the initial formulation.

$$d_{\text{qp}}(\lambda, x) = \|x^0 - \tilde{x}^0\|^2 + \sum_{l=1}^{L-1} \lambda_l \left(x^l\right)^T \left(x^l - \left(W^l x^{l-1} + b^l\right)\right)$$

$$= \left(x^0\right)^T x^0 - 2\left(x^0\right)^T \tilde{x}^0 + \|\tilde{x}^0\|^2 + \sum_{l=1}^{L-1} \lambda_l \left(\left(x^l\right)^T x^l - \left(x^l\right)^T W^l x^{l-1} - \left(x^l\right)^T b^l\right)$$

$$= \underbrace{\sum_{l=0}^{L-1} \lambda_l \left(x^l\right)^T x^l - \sum_{l=1}^{L-1} \lambda_l \left(x^l\right)^T W^l x^{l-1}}_{\text{quadratic term}} \underbrace{-2\left(x^0\right)^T \tilde{x}^0 - \sum_{l=1}^{L-1} \lambda_l \left(x^l\right)^T b^l + \|\tilde{x}^0\|^2}_{\text{linear and constant terms}}$$

From the quadratic term we can identify now the blocks of $M^\lambda$

| | | |
|---|---|---|
| diagonal: $M_{l,l}^\lambda = \lambda_l E_l$ | | $l = 0, \ldots, L-1,$ |
| sub-diagonal: $M_{l+1,l}^\lambda = \left(M_{l,l+1}^\lambda\right)^T = -\frac{1}{2}\lambda_l W^l$ | | $l = 0, \ldots, L-2.$ |

$\square$

**Theorem 1.** *Let $W^1, \ldots, W^{L-1}$ be the weights of an arbitrary pre-trained neural net and $\|W\|$ the spectral norm of an arbitrary matrix. Then the following two conditions for $\lambda$ provide correspondingly a sufficient and a necessary criterion for the matrix $M^\lambda$ to be positive semi-definite.*

$$\text{(suf. condition)} \qquad \lambda_1 \leq \frac{2\lambda_0}{\|W^1\|^2} \text{ and } \lambda_l \leq \frac{\lambda_{l-1}}{\|W^l\|^2} \qquad \text{for } l = 2, \ldots, L-1 \qquad (8)$$

$$\text{(nec. condition)} \qquad \lambda_l \leq \frac{4\lambda_{l-1}}{\|W^l\|^2} \qquad \text{for } l = 1, \ldots, L-1 \qquad (9)$$

*Further, we define $\underline{\lambda}$ and $\bar{\lambda}$ that correspondingly satisfy conditions (8) and (9) with equality:*

$$\underline{\lambda}_l = 2 \prod_{k=1}^{l} \frac{1}{\|W^k\|^2}, \quad \bar{\lambda}_l = 4^l \prod_{k=1}^{l} \frac{1}{\|W^k\|^2}.$$

*In case with a single hidden layer $M^\lambda$ with $\lambda = \bar{\lambda}$ from (9) is guaranteed to be positive-semi definite.*

*Proof.* Let the assumptions hold and $x$ be an arbitrary vector from $\mathbb{R}^n$. First we prove the *sufficient condition* by deriving a lower bound on $x^T M^\lambda x$ that is non-negative if (8) holds.

$$x^T M^\lambda x = \sum_{l=0}^{L-1} \lambda_l \|x^l\|^2 - \sum_{l=1}^{L-1} \lambda_l \left(x^l\right)^T W^l x^{l-1}$$

$$= \frac{\lambda_0}{2} \|x^0\|^2 + \frac{\lambda_{L-1}}{2} \|x^{L-1}\|^2$$

$$+ \sum_{l=1}^{L-1} \frac{\lambda_l}{2} \|x^l\|^2 - \lambda_l \left(x^l\right)^T W^l x^{l-1} + \frac{\lambda_{l-1}}{2} \|x^{l-1}\|^2$$

$$= \frac{\lambda_0}{2} \|x^0\|^2 + \frac{\lambda_{L-1}}{2} \|x^{L-1}\|^2$$

$$+ \sum_{l=1}^{L-1} \frac{\lambda_l}{2} \|x^l\|^2 - \lambda_l \left(x^l\right)^T W^l x^{l-1} + \frac{\lambda_l}{2} \|W^l x^{l-1}\|^2$$

$$+ \sum_{l=1}^{L-1} \frac{\lambda_{l-1}}{2} \|x^{l-1}\|^2 - \frac{\lambda_l}{2} \|W^l x^{l-1}\|^2$$

$$= \frac{\lambda_0}{2} \|x^0\|^2 + \frac{\lambda_{L-1}}{2} \|x^{L-1}\|^2 + \sum_{l=1}^{L-1} \frac{\lambda_l}{2} \|x^l - W^l x^{l-1}\|^2$$

$$+ \sum_{l=1}^{L-1} \frac{\lambda_{l-1}}{2} \|x^{l-1}\|^2 - \frac{\lambda_l}{2} \|W^l x^{l-1}\|^2$$

$$= \frac{\lambda_{L-1}}{2} \|x^{L-1}\|^2 + \sum_{l=1}^{L-1} \frac{\lambda_l}{2} \|x^l - W^l x^{l-1}\|^2 +$$

$$+ \left(\lambda_0 \|x^0\|^2 - \frac{\lambda_1}{2} \|W^1 x^0\|^2\right) + \sum_{l=2}^{L-1} \frac{\lambda_{l-1}}{2} \|x^{l-1}\|^2 - \frac{\lambda_l}{2} \|W^l x^{l-1}\|^2$$

$$\geq \frac{\lambda_{L-1}}{2} \|x^{L-1}\|^2 + \sum_{l=1}^{L-1} \frac{\lambda_l}{2} \|x^l - W^l x^{l-1}\|^2 +$$

$$+ \frac{1}{2} \left(2\lambda_0 - \lambda_1 \|W^1\|^2\right) \|x^0\|^2 + \frac{1}{2} \sum_{l=2}^{L-1} \left(\lambda_{l-1} - \lambda_l \|W^l\|^2\right) \|x^{l-1}\|^2$$

where we applied the sub-multiplicativity property of the spectral norm, i.e. $\|W^l x^{l-1}\| \leq \|W^l\|\|x^{l-1}\|$, to obtain the last inequality. We see that under the assumption (8) on $\lambda$ and $W$'s it holds that

$$\lambda_1 \|W^1\|^2 \leq 2\lambda_0 \text{ and } \lambda_l \|W^l\|^2 \leq \lambda_{l-1} \text{ for } l = 2, \ldots, L-1$$

and the lower bound on $x^T M^\lambda x$ in the last line is a sum of non-negative terms meaning that $x^T M^\lambda x \geq 0$ for all $x \in \mathbb{R}^n$.

To prove the *necessary condition* consider for each $l = 1, \ldots, L - 1$ a special vector $\tilde{x}$ (we don't explicitly label it as dependent on $l$ to avoid overloaded notation) which is everywhere zero except

$$\tilde{x}^{l-1} := \arg \max_{x \in \mathbb{R}^{n_{l-1}}} \frac{\|W^l x\|}{\|x\|} \text{ and } \tilde{x}^l := \frac{1}{2} W^l x^{l-1}.$$

For $M^\lambda$ in order to be positive semi-definite it has to satisfy

$$0 \leq \tilde{x}^T M^\lambda \tilde{x} = \begin{pmatrix} \tilde{x}^{l-1} \\ \tilde{x}^l \end{pmatrix}^T \begin{pmatrix} \lambda_{l-1} E_{l-1} & -\frac{1}{2} \lambda_l (W^l)^T \\ -\frac{1}{2} \lambda_l W^l & \lambda_l E_l \end{pmatrix} \begin{pmatrix} \tilde{x}^{l-1} \\ \tilde{x}^l \end{pmatrix}$$

$$= \lambda_{l-1} \|\tilde{x}^{l-1}\|^2 - \lambda_l (\tilde{x}^l)^T W^l \tilde{x}^{l-1} + \lambda_l \|\tilde{x}^l\|^2$$

$$= \lambda_{l-1} \|\tilde{x}^{l-1}\|^2 - \frac{1}{4} \lambda_l \|W^l \tilde{x}^{l-1}\|^2 = \left( \lambda_{l-1} - \frac{1}{4} \lambda_l \|W^l\|^2 \right) \|\tilde{x}^{l-1}\|^2$$

which results in the necessary condition (9) as stated above. It remains to prove the sufficiency of (9) if the considered network contains one hidden layer. For that we can reuse the last computation and obtain now for an arbitrary $x \in \mathbb{R}^n$ that

$$x^T M^{\bar{\lambda}} x = \lambda_0 \|x^0\|^2 - \lambda_1 (x^1)^T W^1 x^0 + \lambda_1 \|x^1\|^2$$

$$= \lambda_0 \|x^0\|^2 - \frac{1}{4} \lambda_1 \|W^1 x^0\|^2 + \lambda_1 \|\frac{1}{2} W^1 x^0 - x^1\|^2$$

$$\geq \left( \lambda_0 - \frac{1}{4} \lambda_1 \|W^1\|^2 \right) \|x^0\|^2 + \lambda_1 \|\frac{1}{2} W^1 x^0 - x^1\|^2.$$

We see that the last term remains non-negative in case of $\lambda_1 = \frac{4\lambda_0}{\|W^1\|^2}$ for all $x$. $\qquad\square$

**Theorem 2.** *For $k = 0, 1, \ldots$ let $c_k$ be the propagation gap as defined in (7) and $d_k^2$ the optimal objective function value we obtain after solving QPRel in the $k$'th iteration of Algorithm 1. Assume that there exists $\alpha \in [0, 1]$ such that for all $k$ the following holds.*

$$\frac{c_k}{d_k^2} \leq 1 - \alpha^2 \text{ or equivalently } \frac{\|x^0 - x_{qp}^0\|}{d_k} \geq \alpha \tag{10}$$

*where $x^0$ is the anchor point and $x_{qp}$ is the optimal solution of QPRel in $k$'s iteration (we omit the iteration index $k$ on $x$'s). Then for all $k$*

$$d_{k+1}^2 \leq 2(1 - \alpha) d_k^2, \quad c_k \leq (2(1 - \alpha))^{k+1} d_0^2. \tag{11}$$

*Furthermore, if $\alpha > \frac{1}{2}$ then the number of iterations $\bar{k}$ such that Algorithm 1 terminates with $c_{\bar{k}} \leq c_{tol}$ is bounded by*

$$\bar{k} \leq \frac{\log c_{tol} - 2 \log d_0}{\log 2(1 - \alpha)} - 1. \tag{12}$$

*Proof.* First, we prove that given (10) it holds that $d_{k+1}^2 \leq 2(1 - \alpha) d_k^2$. We denote the anchor point in iteration $k$ as $x^0$ and the optimal solution for QPRel with that anchor point as $x_{qp}$. The corresponding objects in the next iteration $k + 1$ are denoted by $y^0$ and $y_{qp}$. Then according to the choice of the next anchor point (see Algorithm 1, line 8)

$$y^0 = x^0 + d_k \frac{x_{qp}^0 - x^0}{\|x_{qp}^0 - x^0\|} \Rightarrow \|y^0 - x_{qp}^0\| = d_k - \|x^0 - x_{qp}^0\| \quad \text{(compare to Figure 2)} \tag{13}$$

and it follows that

$$d_{k+1}^2 = \min_{x:(5),(6)} \|y^0 - x^0\|^2 + c(x, \lambda) \leq \|y^0 - x_{qp}^0\|^2 + c(x_{qp}, \lambda)$$

$$\underset{(13)}{=} \left( d_k - \|x^0 - x_{qp}^0\| \right)^2 + c_k \underset{(10)}{\leq} (1 - \alpha)^2 d_k^2 + (1 - \alpha^2) d_k^2 = 2(1 - \alpha) d_k^2.$$

Now, applying this relation on every two subsequent iterations up to the $k$'s we obtain (here we use $1 - \alpha^2 = (1 + \alpha)(1 - \alpha) \leq 2(1 - \alpha)$ to get the last inequality)

$$c_k \underset{(10)}{\leq} (1 - \alpha^2)d_k^2 \leq (1 - \alpha^2)(2(1 - \alpha))^k d_0^2 \leq (2(1 - \alpha))^{k+1} d_0^2.$$

It remains to prove the last inequality (12). Note that for $1 \geq \alpha > \frac{1}{2}$ the derived upper bound on $c_k$ is monotonically decreasing and converges to 0 as $k \to \infty$ since $0 \leq 2(1 - \alpha) < 1$. Therefore, to find the first $k$ such that the upper bound on $c_k$ becomes smaller or equal $c_{\text{tol}}$ we have to solve the equation $c_{\text{tol}} = (2(1 - \alpha))^{k+1} d_0^2$ for $k$ resulting in the upper bound on $k$ as claimed in (12).

$\square$

## D  TABLES

Table 6: MOSEK parameters we use to run SDPRel and their default values

| Parameter [1] | New value | Default value |
| --- | --- | --- |
| MSK_IPAR_NUM_THREADS | 1 | 0 |
| MSK_DPAR_INTPNT_CO_TOL_MU_RED | $10^{-4}$ | $10^{-8}$ |
| MSK_DPAR_INTPNT_CO_TOL_REL_GAP | $10^{-4}$ | $10^{-8}$ |
| MSK_DPAR_INTPNT_CO_TOL_INFEAS | $10^{-6}$ | $10^{-12}$ |
| MSK_DPAR_INTPNT_CO_TOL_DFEAS | $10^{-4}$ | $10^{-8}$ |
| MSK_DPAR_INTPNT_CO_TOL_PFEAS | $10^{-4}$ | $10^{-8}$ |

---

[1]`https://docs.mosek.com/9.0/pythonapi/parameters.html` contains the full list of parameters including their description.

Table 7: Results for train data, $l_2$-perturbations

| Setting | | | | MedRelDiff | $\epsilon$ to hit 50% | VerRatio | MedRelDiff |
|---|---|---|---|---|---|---|---|
| Data | L/N | NrPts | Method | to QPRel (%) | LB-verified | worst (%) | UB-LB (%) |
| MNIST | 1/100 | 1086 | QPRel-LB | – | **0.246** | **82.8** | **13.4** |
| | 1/100 | 1086 | CROWN+FGSM | **+82.4** | 0.129 | 8.1 | 1033.7 |
| | 2/100 | 965 | QPRel-LB | – | **1.174** | **99.3** | **<0.01** |
| | 2/100 | 965 | ConvAdv+FGSM | **+30.1** | 0.891 | 23.3 | 277.9 |
| | 1/300 | 1142 | QPRel-LB | – | **0.243** | **94.0** | **5.5** |
| | 1/300 | 1142 | CROWN+FGSM | **+149.1** | 0.101 | 10.9 | 627.8 |
| | 1/50 | 1180 | QPRel-LB | – | **0.872** | **82.4** | **14.3** |
| | 1/50 | 1180 | CROWN+FGSM | **+67.0** | 0.506 | 8.8 | 419.4 |
| F-MNIST | 1/100 | 1083 | QPRel-LB | – | **0.252** | **81.9** | **17.2** |
| | 1/100 | 1083 | CROWN+FGSM | **+49.1** | 0.159 | 15.9 | 678.8 |
| | 2/100 | 924 | QPRel-LB | – | **0.332** | **83.0** | **20.3** |
| | 2/100 | 924 | ConvAdv+FGSM | **+68.0** | 0.205 | 22.2 | 375.1 |
| | 1/300 | 1094 | QPRel-LB | – | **0.183** | **78.8** | **23.1** |
| | 1/300 | 1094 | CROWN+FGSM | **+56.6** | 0.122 | 12.8 | 902.2 |

Table 8: Results for test data, $l_2$-perturbations

| Setting | | | | MedRelDiff | $\epsilon$ to hit 50% | VerRatio | MedRelDiff |
|---|---|---|---|---|---|---|---|
| Data | L/N | NrPts | Method | to QPRel (%) | LB-verified | worst (%) | UB-LB (%) |
| MNIST | 1/100 | 1137 | QPRel-LB | – | **0.268** | **81.9** | **14.6** |
| | 1/100 | 1137 | CROWN+FGSM | **+84.6** | 0.146 | 6.7 | 1097.2 |
| | 2/100 | 1028 | QPRel-LB | – | **1.251** | **99.4** | **0.0** |
| | 2/100 | 1028 | ConvAdv+FGSM | **+30.2** | 0.959 | 23.6 | 281.8 |
| | 1/300 | 1166 | QPRel-LB | – | **0.267** | **93.9** | **5.9** |
| | 1/300 | 1166 | CROWN+FGSM | **+154.6** | 0.111 | 7.0 | 658.5 |
| | 1/50 | 1212 | QPRel-LB | – | **0.937** | **80.0** | **15.2** |
| | 1/50 | 1212 | CROWN+FGSM | **+68.5** | 0.562 | 7.4 | 421.8 |
| F-MNIST | 1/100 | 1061 | QPRel-LB | – | **0.258** | **79.8** | **19.2** |
| | 1/100 | 1061 | CROWN+FGSM | **+50.3** | 0.164 | 13.7 | 703.0 |
| | 2/100 | 949 | QPRel-LB | – | **0.355** | **79.9** | **21.3** |
| | 2/100 | 949 | ConvAdv+FGSM | **+71.6** | 0.213 | 19.3 | 375.2 |
| | 1/300 | 1069 | QPRel-LB | – | **0.194** | **77.0** | **24.7** |
| | 1/300 | 1069 | CROWN+FGSM | **+59.3** | 0.122 | 10.6 | 890.9 |

Table 9: Results for train data, $l_\infty$-perturbations

| Setting | | | | MedRelDiff | $\epsilon$ to hit 50% | VerRatio | MedRelDiff |
|---|---|---|---|---|---|---|---|
| Data | L/N | NrPts | Method | to QPRel (%) | LB-verified | worst (%) | UB-LB (%) |
| MNIST | 1/100 | 1086 | QPRel-LB | – | 0.010 | 5.8 | 1478.1 |
| | 1/100 | 1086 | CROWN+FGSM | **+33.4** | 0.010 | **13.6** | **882.5** |
| | 2/100 | 965 | QPRel-LB | – | 0.057 | 5.3 | 2484.0 |
| | 2/100 | 965 | ConvAdv+FGSM | **+16.8** | 0.057 | **32.4** | **220.4** |
| | 1/300 | 1142 | QPRel-LB | – | 0.011 | 3.6 | 1691.4 |
| | 1/300 | 1142 | CROWN+FGSM | **+104.3** | 0.011 | **13.0** | **575.1** |
| | 1/50 | 1180 | QPRel-LB | – | 0.033 | 1.6 | 3344.0 |
| | 1/50 | 1180 | CROWN+FGSM | **+2.2** | 0.033 | **16.0** | **286.8** |
| F-MNIST | 1/100 | 1083 | QPRel-LB | – | 0.010 | 6.9 | 1520.8 |
| | 1/100 | 1083 | CROWN+FGSM | **+5.5** | 0.010 | **17.8** | **599.5** |
| | 2/100 | 924 | QPRel-LB | – | 0.013 | 3.6 | 1855.8 |
| | 2/100 | 924 | ConvAdv+FGSM | **+17.1** | 0.013 | **33.1** | **341.6** |
| | 1/300 | 1094 | QPRel-LB | – | 0.007 | 3.0 | 1579.8 |
| | 1/300 | 1094 | CROWN+FGSM | **+11.0** | 0.007 | **14.2** | **741.8** |

Table 10: Results for test data, $l_\infty$-perturbations

| Setting | | | | MedRelDiff | $\epsilon$ to hit 50% | VerRatio | MedRelDiff |
|---|---|---|---|---|---|---|---|
| Data | L/N | NrPts | Method | to QPRel (%) | LB-verified | worst (%) | UB-LB (%) |
| MNIST | 1/100 | 1137 | QPRel-LB | – | 0.011 | 7.2 | 1439.0 |
| | 1/100 | 1137 | CROWN+FGSM | **+33.6** | 0.011 | **10.5** | **946.5** |
| | 2/100 | 1028 | QPRel-LB | – | **0.115** | 5.2 | 2480.9 |
| | 2/100 | 1028 | ConvAdv+FGSM | **+16.5** | 0.058 | **28.7** | **218.6** |
| | 1/300 | 1166 | QPRel-LB | – | 0.011 | 2.4 | 1704.3 |
| | 1/300 | 1166 | CROWN+FGSM | **+105.7** | 0.011 | **8.1** | **610.7** |
| | 1/50 | 1054 | QPRel-LB | – | 0.037 | 1.8 | 3462.1 |
| | 1/50 | 1054 | CROWN+FGSM | +2.3 | 0.037 | 15.5 | 280.8 |
| | 1/50 | 1054 | SDPRel+FGSM | **-54.5** | **0.074** | **51.1** | **68.1** |
| F-MNIST | 1/100 | 1061 | QPRel-LB | – | 0.011 | 6.8 | 1520.9 |
| | 1/100 | 1061 | CROWN+FGSM | **+5.5** | 0.011 | **19.3** | **605.4** |
| | 2/100 | 949 | QPRel-LB | – | 0.012 | 2.5 | 1938.1 |
| | 2/100 | 949 | ConvAdv+FGSM | **+18.2** | 0.012 | **28.6** | **344.7** |
| | 1/300 | 1069 | QPRel-LB | – | 0.007 | 3.9 | 1543.1 |
| | 1/300 | 1069 | CROWN+FGSM | **+12.6** | 0.007 | **12.9** | **736.9** |

