# OpenReview forum: "Efficient Bi-Directional Verification of ReLU Networks via Quadratic Programming"
_ICLR.cc/2020/Conference — Reject_

### Official Review · AnonReviewer2 · 2019-10-22
**Official Blind Review #2**

**Rating:** 3

**Review:**

* Summary:
This paper introduces an encoding of the bounds on Neural Networks based on a (non-convex) quadratic program. The method covers both L_inf and L_2 perturbations, and the optimization problem posed are solved using Gurobi.

The authors formulate the ReLU as a quadratic constraint , and then relax the problem by taking it's Lagrangian dual. This results in the standard properties from taking a lagrangian dual: any choice of dual variables will produce a lower bound.
For an appropriate choice of dual variables, the lagrangian problem is convex and the authors give a way of choosing dual variables that guarantees this convexity such that the bound can be computed.

A method is proposed to find upper bound on the verified region (which can also be understood as just finding an incorrectly classified sample, that should ideally be the closest to the original point), based on iterating the solving of the QP.

Comments:
- Encoding of the ReLU as a quadratic constraint as done in (2) is not novel, as it was done before by Dvijotham et al. (UAI 2019) or Raghunathan et al. (NeurIPS 2018). The Lagrangian relaxation that is then done is to the best of my knowledge different than any one introduced before.

- The problem solved is also different to most of the literature: rather than verifying robustness for a certain radius, this attempts to find the maximum radius being robust, which provide more information.

- I'm confused at the upper bound finding method. The solution of the QP will not necessarily respect the constraints of forward propagation of the network so if you just consider the variables corresponding to the input, the resulting output may not necessarily a violation. Also, I don't understand the motivation for why repeatedly solving the QP will lead to a good violation on the decision boundary. I know that the paper says that "analytic investigation of this algorithm including a convergence proof remains future work." but at the moment there is not even an intuition for why it might be a good idea. By the second iteration, there is no notion of the reference point around which safety is computed so it's not sure how the closest violation would be found.

- The network tested are extremely small, even by formal verification of neural network standards, which makes it hard to appreciate the impact of the method and makes me question the applicability of the method. Is it because QP are more complex to solve than LPs?
- It is also a little bit problematic to give results as ratio of improvements over the lower bound on radius, when most of the network used are non robust, given that those networks have extremely low verified radius, so the relative difference will look inflated.

- The reporting of the verification ratio as a function of the perturbation radius is an interesting measure that I think is very benificial to making the point but I think it should be better explained as it took me a long time to get the point. The experiment section in general is quite confuse and hard to parse, having to jump around quite a lot to get what the author meant.

- FGSM is a very very weak baseline for the use that is employed here. By construction, it doesn't look for the smallest violation, is not iterative, and produce perturbations at the limit of the attacked budget.

- The paper takes the opportunity to say that their method is 2000x faster than SDP based method but not that they are 10x to 100x slower than CROWN (outside from the appendix). It's better to report results clearly than only trying to show the good points of the algorithm. The bounds obtained are tighter than those resulting from Crown so it might be a worthwile tradeoff to make.

* Probably worth discussing / comparing to:
Provable Certificates for Adversarial Examples: Fitting a Ball in the Union of Polytopes, Jordan et al. (ICML 2019)

* Typos and minor comments:
- Adjust the label for Figure 2
- "Guarantied" on page 4
- top of page 8 "VerRation"




**Experience Assessment:**

I have published one or two papers in this area.

**Review Assessment: Checking Correctness Of Derivations And Theory:**

I assessed the sensibility of the derivations and theory.

**Review Assessment: Checking Correctness Of Experiments:**

I carefully checked the experiments.

**Review Assessment: Thoroughness In Paper Reading:**

I read the paper thoroughly.

---

> ### Author Response · Authors · 2019-11-15
> **Revised Version and Reply - Part 1**
>
> Thank you for your review. Indeed, one of the main advantages of our approach is that it does not require a certain radius as input but aims to find the largest one - this makes our approach also very different to other approaches based on LP or SDP relaxations.
>
>
> 1. Upper bounds: Please see Section 4 for an updated and extended discussion of our method to compute upper bounds on DtDB, which can be summarized as follows:
>
> Idea: In each step we verify a certain neighborhood around the current anchor point $x^0$ and then expand the verified region further. For that we choose the next anchor point as follows.
> (i) Choice of direction: we go from $x^0$ towards the solution of QPRel since, if the QP relaxation is tight, its solution should be close to an optimal solution of DtDB which is the closest adversarial point to $x^0$.
> (ii) Choice of step size: we know that there are no adversarial points within the ball of the verified radius $d$ around the anchor, so every step size smaller than $d$ would be unnecessary small. On the other hand, if we proceed with a new anchor point that is strictly farther away than $d$, we might miss an adversarial point lying close to the boundary of the $d$-ball around $x^0$. Therefore, we choose the next anchor point to be on the boundary of the currently verified region, so that every $\epsilon$-ball that we manage to verify around the new point would add to the overall robust set.
>
> Termination: The algorithm terminates as soon as the propagation gap $c(x,\lambda)$ becomes small enough or the anchor point gets misclassified. Note that $c(x, \lambda)=0$ means that the solution $x$ provides the optimal objective function value of the DtDB problem (see Lemma 1) and, thus, an adversarial example. However, the termination condition $c\le c_{\text{tol}}$ from Algorithm 1, line 3 cannot ensure that the optimal point $x_{\text{qp}}^0$ from the last iteration belongs to a different class. Therefore, if we stop with $c\le c_{\text{tol}}$ and the second termination condition is not satisfied, we take an additional step on the boundary of the ball of radius $d_{\text{qp}}(1+\delta)$, where $d_{\text{qp}}$ is the verified radius and $\delta$ is a tiny offset. Additionally, we check in each step whether the next anchor point is already misclassified before the condition $c\le c_{\text{tol}}$ is reached (this can happen in a multi-class setting; and is indeed observed frequently). This means that the sequence of anchor points converges towards the boundary and then, if no adversarial point was found yet, makes a step across the boundary using a positive, small $\delta$.
>
> Convergence: We also included a result for the convergence rate of Algorithm 1.

---

> > ### Author Response · Authors · 2019-11-15
> > **Revised Version and Reply - Part 2**
> >
> > 2. Network size: We considered deeper networks with 50 nodes per hidden layer and their number varying from 1 to 5. The resulting relaxation for deeper nets (number hidden layers > 2) was shown to be looser than for small networks resulting in worse *lower* bounds compared to CROWN. The upper bounds computed by QPRel were consistently better across different model choices. See Table 1 for the updated results.
> >
> > A probable reason for this decrease in performance of the lower bound is the choice of $\lambda$ that we perform along one line. For many layers the space of $\lambda\in\mathbb{R}^L$ becomes higher dimensional so that a more sophisticated search in this space for a good $\lambda$ is required. Note that according to Theorem 1 in the setting of only 1 hidden layer we are solving provably the tightest convex QP relaxation possible, which is not guaranteed for deeper architectures.
> >
> >
> > 3. Relative vs. absolute improvements: We added the average (absolute) difference between the lower bounds to the tables. Furthermore, we considered normally and robustly trained networks for each architecture.
> >
> > We decided to use the relative differences additionally to the absolute ones because it is hard to judge whether an absolute improvement of $\Delta\epsilon$ of the lower bound is significant or not without knowing the true DtDB or the ratio between the two bounds. Our empirical investigations clearly show (see Fig. 3a in the initial submission) that samples might have very different DtDB, some are close to the decision boundary and some are far away from it. On the other side we consider an improvement of 0.01% always as non-significant and an improvement of 100% as significant independently of the actual DtDB.
> >
> >
> > 4. Stronger attacks: We conducted additional experiments with FGSM replaced by 200-steps PGD (starting from the anchor point, no random sampling). Experiments show that QPRel-UB outperforms them in $l_2$ setting on all architectures allowing for more samples to be verified as non-robust. Results are included into the updated version of the paper (see Table 1).
> >
> >
> > 5. Speed-up: We have adjusted the statements in the updated version of the paper.
> >
> >
> > 6. Related work: We have included a discussion of the work of Jordan et al. (2019) in Section 4. Thank you for pointing it out!

---

### Official Review · AnonReviewer3 · 2019-10-23
**Official Blind Review #3**

**Rating:** 8

**Review:**

This paper proposes a method to compute the distance to the decision boundary for a given network, where the network is composed of linear layers followed by a RELU activation. The authors provide a lower bound and an upper bound for the distance of a sample from the decision boundary. The lower bound is obtained as the solution to a quadratic program, which in turn is obtained by relaxing the original optimization problem. The relaxation is obtained by decomposing the RELU condition to a set of 3 constraints (eqn 2). The authors also provide conditions under which the quadratic program stays convex.

The paper is clearly written. The method is useful to verify robustness of neural networks. The experiments show the improvement of the proposed method over existing certificates.

While the lower bound is theoretically justified, I did not see any guarantees for the upper bound. I am not referring to a convergence proof here, but simply a guarantee that the value returned by Algorithm 1 is indeed an upper bound. Algorithm 1 does not verify whether the point returned belongs to a different class. It would also be helpful to provide intuition for the iterative procedure to compute the upper bound.

Minor comments:
1) Line 9 is Algorithm 1: x^qp0 should be x^qp (no 0)
2) What is L/N in Table 1?

**Experience Assessment:**

I do not know much about this area.

**Review Assessment: Checking Correctness Of Derivations And Theory:**

I assessed the sensibility of the derivations and theory.

**Review Assessment: Checking Correctness Of Experiments:**

I carefully checked the experiments.

**Review Assessment: Thoroughness In Paper Reading:**

I read the paper thoroughly.

---

> ### Author Response · Authors · 2019-11-15
> **Revised Version and Reply**
>
> Thank you for your review. We have updated and extended the discussion of our method to compute upper bounds on DtDB (see Section 4 in the paper and summary below).
>
> ** Upper bounds **
>
> Idea: In each step we verify a certain neighborhood around the current anchor point $x^0$ and then expand the verified region further. For that we choose the next anchor point as follows.
> (i) Choice of direction: we go from $x^0$ towards the solution of QPRel since, if the QP relaxation is tight, its solution should be close to an optimal solution of DtDB which is the closest adversarial point to $x^0$.
> (ii) Choice of step size: we know that there are no adversarial points within the ball of the verified radius $d$ around the anchor, so every step size smaller than $d$ would be unnecessary small. On the other hand, if we proceed with a new anchor point that is strictly farther away than $d$, we might miss an adversarial point lying close to the boundary of the $d$-ball around $x^0$. Therefore, we choose the next anchor point to be on the boundary of the currently verified region, so that every $\epsilon$-ball that we manage to verify around the new point would add to the overall robust set.
>
> Termination: The algorithm terminates as soon as the propagation gap $c(x,\lambda)$ becomes small enough or the anchor point gets misclassified. Note that $c(x, \lambda)=0$ means that the solution $x$ provides the optimal objective function value of the DtDB problem (see Lemma 1) and, thus, an adversarial example. However, the termination condition $c\le c_{\text{tol}}$ from Algorithm 1, line 3 cannot ensure that the optimal point $x_{\text{qp}}^0$ from the last iteration belongs to a different class. Therefore, if we stop with $c\le c_{\text{tol}}$ and the second termination condition is not satisfied, we take an additional step on the boundary of the ball of radius $d_{\text{qp}}(1+\delta)$, where $d_{\text{qp}}$ is the verified radius and $\delta$ is a tiny offset.
>
> Additionally, we check in each step whether the next anchor point is already misclassified before the condition $c\le c_{\text{tol}}$ is reached (this can happen in a multi-class setting; and is indeed observed frequently). We empirically verified that all points obtained this way are indeed true adversarials. This means that the sequence of anchor points converges towards the boundary and then, if no adversarial point was found yet, makes a step across the boundary using a positive, small $\delta$.
>
> Convergence: We also included a result for the convergence rate of Algorithm 1 under an assumption that the propagation gap $c(x,\lambda)$ is not too large with respect to the optimal objective function value for the QPRel problem solved in each step.
>
> ** Minor comments **
>
> 1. We have rewritten Algorithm 1 in the updated version of the paper to fix typos and make it easier to read.
>
> Note that we use the upper case index on $x$ throughout the paper to indicate the layer, so $x^0$ is the sample in the input layer and $x^L=f(x^0)$ are values of the last layer, when $x^0$ is propagated through the net. The intermediate activations are denoted by $x^l$. We use this notation also in Algorithm 1 and we actually omit the index identifying the number of the current iteration. So also here $x^0$ denotes the part of the full $x$ corresponding to the input layer and we return at the end $x_{adv}^0 \leftarrow  x^0$ as an adversarial point (alternatively we could write $x_{adv} \leftarrow  x$).
>
> 2. L, N and R denote the network architecture. L is the number of hidden layers, N is the number of hidden neurons per hidden layer (same for each hidden layer) and R encodes whether the network was trained robustly (if R is present) or normally (if there is no R).

---

### Official Review · AnonReviewer1 · 2019-10-25
**Official Blind Review #1**

**Rating:** 6

**Review:**

This paper proposed to use a convex QP relaxed formulation to solve the neural network verification problem, and demonstrated its effectiveness on a few small networks (1-2 hidden layers) on MNIST and Fashion-MNIST datasets.

There are several benefits the proposed methods: they are technically tighter relaxations of ReLU neurons and empirically the authors show they perform well in L2 norm (but not L infinity norm, unfortunately); solving this formulation does not require to know pre-activation bounds of hidden neurons; also the convexity of the QP problem needs to be determined only once for a model, rather than once per example. Although the QP relaxation of ReLU neuron is not new and has been used in Raghunathan et al., 2018b, they solve the problem as a SDP rather than convex QP. SDP is tighter than the convex QP formulation used in this paper, however is much slower.

Issues and Questions:

1. The concept of "bi-direction verification" is not new, since finding an upper bound is basically finding adversarial examples. Many previous papers have been using PGD based attacks to obtain the upper bound. Convex relaxation based verification methods like CROWN can also be used for generating adversarial examples, and it is called "Interval attack", which is demonstrated in [1][2]. Claiming this is the first "bi-direction robustness verification technique" is not accurate.

2. The use of FGSM as an upper bound is inappropriate, as FGSM is known to be a very weak attack. Replacing it with a multi-step PGD attack is necessary. Using a stronger attack will also close the gap between upper and lower bound. Also, compare the upper bound found by PGD with QPRel-LB and update Figure 3(a). If a stronger attack like PGD is used, I think for larger norms CROWN+PGD in Figure 3(c) should be able to verify almost all examples.

3. The models used in Table 1 is trained using a L2 perturbation of epsilon=0.1. This epsilon value is too small for L2 norm. In page 22 (last page of appendix in arxiv version) of [3], you can find the they conduct L2 robustness training but at a much larger epsilon value (eps=1.58). Sine the authors did not use these standard epsilon setting, my concern is that does the proposed method works at larger L2 epsilon?

4. Some experiments on larger and deeper networks are necessary; especially, it is interesting to see how CROWN and the proposed method scale to deeper networks. The presented experiments only include networks with 1 and 2 hidden layers, which is insufficient. A new experiment with number of hidden neurons per layer kept (say 50) and increase the depth from 2 to 10 will be very helpful.

5. The main claim of the paper in Introduction needs to be made clearer, especially the primary strength of the proposed algorithm is in L2 norm, and it does not seem to outperform CROWN in L infinity norm setting.

Further improvements and potential directions:

1. In the proposed method, the authors relaxed ReLU neurons using quadratic programming. This relaxation does not require to computing bounds for the neuron activation values. However, I think it is possible to include neuron activation upper and lower bounds as constraints of the QP problem (adding them as constraints like l <= x <= u in Eq. QPRel). This will make the bounds tighter. The per-neuron lower and upper bounds can be obtained using CROWN efficiently, so there is no too much computation cost.

2. Improving the scalability of QP relaxation is another challenge. CROWN can be implemented efficiently on GPUs [4]. For QP relaxations, this can possibly be done by transforming QP solving into a computation graph that can be executed efficiently on GPUs (this is a potential future work directions and I do not expect the authors to address them during the discussion period).

Overall I am positive with this paper, however before accepting it I think the authors should at least make their claims clearer (the relaxation performs well mainly in L2 norm, and the concept of "bi-directional verification" is also not entirely new), replacing FGSM by a 200-step PGD and compare the upper bound found by PGD with QPRel, and test the proposed algorithm in models trained with a larger epsilon (eps=1.58 to align with previous works, if possible) and deeper models.

[1] Wang, S., Chen, Y., Abdou, A., & Jana, S. (2019). Enhancing Gradient-based Attacks with Symbolic Intervals. arXiv preprint arXiv:1906.02282.
[2] Wang, S., Chen, Y., Abdou, A., & Jana, S. (1811). MixTrain: Scalable Training of Verifiably Robust Neural Networks.
[3] Wong, E., Schmidt, F., Metzen, J. H., & Kolter, J. Z. (2018). Scaling provable adversarial defenses. In Advances in Neural Information Processing Systems (pp. 8400-8409).
[4] https://github.com/huanzhang12/RecurJac-and-CROWN


**Experience Assessment:**

I have published in this field for several years.

**Review Assessment: Checking Correctness Of Derivations And Theory:**

I assessed the sensibility of the derivations and theory.

**Review Assessment: Checking Correctness Of Experiments:**

I carefully checked the experiments.

**Review Assessment: Thoroughness In Paper Reading:**

I read the paper thoroughly.

---

> ### Author Response · Authors · 2019-11-15
> **Revised Version and Reply - Part 1**
>
> Thank you for your review. We address your points below.
>
> ** Issues and Questions **
>
> 1. SUMMARY ON "BI-DIRECTIONAL VERIFICATION": We agree that our statement "the first bi-directional robustness verification technique" without an appropriate discussion is unclear. We have added the points discussed below to the updated version of the paper (see Sections 1 and 4).
>
> Discussion:
> It is true that an arbitrary misclassified example provides an upper bound on the distance to the decision boundary (DtDB) and can be used together with a verification method (like CROWN, ConvAdv or SDPRel) to bound the DtDB from both directions. We used exactly this framework to compare with the results obtained from QPRel-UB in the experimental section.
>
> While QPRel uses its solution of a *verification task* $x_{qp}^0$ as an indicator of the direction towards the decision boundary, most of the other attacks (including FGSM, iterative PDG, Carlini-Wagner and the interval attack by Wang et al. (2019)) are gradient-based methods that perform steps towards a solution of an optimization problem constructed to, e.g., maximize the training loss with respect to the label of the anchor point. Even when a bound propagation technique is employed to find an adversarial, there is *not a robustness verification method* being applied.
>
> In short: In contrast to attacks which are inspired by a misclassification task, our methodology emerges from verification. Verify as much as possible until the decision boundary is reached. This idea is exactly the reason why we call QPRel a bi-directional *verification* technique and not verification plus an attack.
>
> Besides this difference in the methodology we have shown the quality of our obtained upper bounds to outperform a 200-steps PGD attack (see Table 1).
>
> We summarize the relation between QPRel (-LB and -UB) and the verification+attack (V+A) framework (e.g. CROWN+PGD or CROWN+interval attack) as follows.
>
> a) The adversarial-free verified region from QPRel is larger than the initially verified neighborhood around the anchor point, since we iteratively proceed solving a verification task around new anchor points leading to the decision boundary (see the hatched region in Figure 2). V+A returns also an initial verified region, but a subsequent attack can provide only a single point/a sequence of single points that are non-adversarial.
>
> b) For QPRel the most closely related attack is Carlini-Wagner as it also works with a relaxation of DtDB, but instead of solving it exactly a gradient-based approach is applied to find a feasible (i.e. misclassified) point that is possibly close to the anchor.
>
> c) As other attacks usually apply an optimization solver (e.g. L-BFGS, Adam or just PGD steps) on non-convex problems there are no theoretical guaranties on their success or convergence rate. In our setting we can solve the initial convex QPRel exactly (allowing us to get valid lower bounds in the first place) and provide a proof for a certain convergence rate of QPRel-UB given that the relaxation is tight enough.
>
> d) QPRel does not need to know the loss used to train the classifier, only the final weights are required. Therefore it is applicable in settings where we get the model only after it was trained. Common attacks (but not all, an exception is e.g. Carlini-Wagner) would have to use a substitute for the loss function and rely on its similarity to the true one in this semi-white box setting. In general, our proposed methodology would allow an attack in every setting where a verification is possible and yields an indication of the direction towards the decision boundary.

---

> > ### Author Response · Authors · 2019-11-15
> > **Revised Version and Reply - Part 2**
> >
> > 2. STRONGER ATTACKS:
> > We conducted additional experiments with FGSM replaced by 200-steps PGD (starting from the anchor point, no random sampling).
> > Experiments show that QPRel-UB still outperforms them in $l_2$ setting on all architectures allowing for more samples to be verified as non-robust.
> > Results are included in the updated version of the paper.
> >
> > 3. LARGER EPSILON + 4. NETWORK ARCHITECTURE:
> > We have rerun the experiments on deeper networks with 50 nodes per hidden layer and their number varying from 1 to 5 each trained normally and robustly with the proposed $\epsilon=1.58$. Experiments have shown that the qualitative difference in results from QPRel and CROWN does not depend on the training procedure as much as on the depth.
> >
> > One sees that for the smaller networks QPRel is always able to verify non-trivial bounds, while the performance on larger networks w.r.t. the lower bound becomes worse than CROWN's. The upper bound computed by QPRel outperforms the competitors in all settings.
> > See the results in Table 1 in the updated version of the paper.
> >
> > A probable reason for the decrease in performance of the lower bound is the choice of $\lambda$ that we perform along one line.
> > For many layers the space of $\lambda\in\mathbb{R}^L$ becomes higher dimensional so that a more sophisticated search in this space for a good $\lambda$ is required. Note that according to Theorem 1 in the setting of only 1 hidden layer we are solving provably the tightest convex QP relaxation possible, which is not guaranteed for deeper architectures.
> >
> > 5. CONTRIBUTIONS OF PAPER:
> > We have adjusted the phrasing accordingly.
> >
> > ** Further improvements and potential directions **
> >
> > 1. We agree that incorporating additional knowledge e.g. in form of activation bounds might improve the performance especially since it would bound the propagation gap $c$. However, to obtain the bounds in the intermediate layers one has to provide the algorithm with some bounds on the perturbations of the input. One of the main advantages of QPRel is that it does *not* rely on any bounds in the input layer.
> >
> > 2. Thank you for pointing that out! This is definitely an interesting direction for future work.

---

### Decision · Program_Chairs · 2019-12-19

**Decision:**

Reject

**Comment:**

This article is concerned with sensitivity to adversarial perturbations. It studies the computation of the distance to the decision boundary from a given sample in order to obtain robustness certificates, and presents an iterative procedure to this end. This is a very relevant line of investigation. The reviewers found that the approach is different from previous ones (even if related quadratic constraints had been formulated in previous works). However, they expressed concerns with the presentation, missing details or intuition for the upper bounds, and the small size of the networks that are tested. The reviewers also mentioned that the paper could be clearer about the strengths and weaknesses of the proposed algorithm. The responses clarified a number of points from the initial reviews. However, some reviewers found that important aspects were still not addressed satisfactorily, specifically in relation to the justification of the approach to obtain upper bounds (although they acknowledge that the strategy seems at least empirically validated), and reiterated concerns about the scalability of the approach. Overall, this article ranks good, but not good enough.